# Games and enculturation: A cross-cultural analysis of cooperative goal structures in Austronesian games

**Sarah M. Leisterer-Peoples**[1,2]*, **Cody T. Ross**[3], **Simon J. Greenhill**[4,5], **Susanne Hardecker**[6], **Daniel B. M. Haun**[1,2]

1 Department of Comparative Cultural Psychology, Max Planck Institute for Evolutionary Anthropology, Leipzig, Germany, 2 Leipzig Research Center for Early Child Development, Universität Leipzig, Leipzig, Germany, 3 Department of Human Behavior, Ecology and Culture, Max Planck Institute for Evolutionary Anthropology, Leipzig, Germany, 4 Department of Linguistic and Cultural Evolution, Max Planck Institute for the Science of Human History, Jena, Germany, 5 ARC Center of Excellence for the Dynamics of Language, ANU College of Asia and the Pacific, Australian National University, Canberra, Australia, 6 SRH University of Applied Health Sciences, Gera, Germany

☯ These authors contributed equally to this work.

* sarah_peoples@eva.mpg.de

**Data Availability Statement:** All relevant data are within the manuscript and the S1 File.

**Funding:** This project was funded by the Max Planck Society for the Advancement of Science

## Abstract

While most animals play, only humans play games. As animal play serves to teach offspring important life-skills in a safe scenario, human games might, in similar ways, teach important culturally relevant skills. Humans in all cultures play games; however, it is not clear whether variation in the characteristics of games across cultural groups is related to group-level attributes. Here we investigate specifically whether the cooperativeness of games covaries with socio-ecological differences across cultural groups. We hypothesize that cultural groups that engage in frequent inter-group conflict, cooperative sustenance acquisition, or that have less stratified social structures, might more frequently play cooperative games as compared to groups that do not share these characteristics. To test these hypotheses, we gathered data from the ethnographic record on 25 ethnolinguistic groups in the Austronesian language family. We show that cultural groups with higher levels of inter-group conflict and cooperative land-based hunting play cooperative games more frequently than other groups. Additionally, cultural groups with higher levels of intra-group conflict play competitive games more frequently than other groups. These findings indicate that games are not randomly distributed among cultures, but rather relate to the socio-ecological settings of the cultural groups that practice them. We argue that games serve as training grounds for group-specific norms and values and thereby have an important function in enculturation during childhood. Moreover, games might server an important role in the maintenance of cultural diversity.

## Introduction

Games are a human invention, but their function might be similar to the function of play in the animal kingdom more generally. As with play, games help children prepare cognitively

(www.mpg.de). The funders had no role in study design, data collection and analysis, decision to publish, or preparation of the manuscript.

**Competing interests:** The authors have declared that no competing interests exist.

and behaviorally for life as successful adults [1]. To assess the possible function of games as training grounds for complex social skills (i.e., shared intentionality [2], or complex coordinated actions), we test if the goal structures of rule-based games are related to key socio-ecological factors across cultural groups on the Austronesian language phylogeny. We focus on one of the most relevant, potentially unique features of the human species: cooperation.

Humans stand out in the natural world for their ability to cooperate with unrelated individuals [3] and organize behavior around shared goals [2]. Such levels of cooperation appear to exceed what can be explained using standard evolutionary models—e.g., based on kinship or reciprocity [4, 5]. An emerging approach to understanding cooperation in humans proposes that cultural group selection [6–9] can explain the high levels of extant cooperation as a function of selection on cultural traits that confer success in inter-group competition—be it direct competition through warfare [10, 11], or indirect competition through processes like differential between-group imitation [12].

Within this framework, cultural variants that serve to increase group-level fitness sufficiently are expected to increase in frequency, at least in contexts where inter-group competition is high [10, 11]. Here, we explore variation in one such cultural variant—the goal structure of rule-based games. While some games emphasize cooperative behavior among members of a group with a shared intention or goal (e.g., in volleyball), others emphasize solitary behaviors (e.g., in knucklebones), or direct competition between individuals (e.g., in chess). We propose three possible scenarios promoting the emergence and maintenance of cooperative goal structures in rule-based games.

The first proposal links the **foraging niche** of a given cultural group with the goal structures of its games. If there are opportunities for mutual benefits to cooperative action [13] for adult foragers, then we would expect selection to favor games with goal structures that emphasize social coordination and cooperativeness.

The second proposal links the **prevalence and type of conflict** in a given cultural group with the goal structure of its games. If there are opportunities for mutual benefits to cooperative action (e.g., in contexts where inter-group warfare is common), then we would again expect selection to favor games with goal structures that emphasize social coordination and cooperativeness.

The third proposal links the **social structure** of a given cultural group with the goal structure of its games. An early study on the distribution of game types across cultural groups indicates that social complexity—operationalized using political integration and the presence or absence of social classes—covaries positively with games of strategy [14]. Additionally, more recent studies suggest that the social structure of cultural groups might be linked with the cooperativeness of the games played therein [15–17]. If the unequal distribution of valuable resources [18] within a society leads to a breakdown of cooperativeness and egalitarian social norms, then we would expect fewer cooperative games in societies with higher levels of social stratification.

We investigate these possible associations by modeling the relationship between the goal structure [19, 20] of games in 25 ethnolinguistic groups within the reconstructed Austronesian language phylogeny [21] and several group-level variables. Specifically, we use data on the goal structure of rule-based games [22] and model these data as a function of the presence or absence of conflict, the extent of land- and water-based hunting in groups, and social stratification [23]. The cultural groups in our sample share a common linguistic [21, 24] and cultural history [23, 25], and are part of one of the largest language families in the world. These ethnolinguistic groups exhibit high levels of cultural diversity [23, 26], and are generally ethnographically well-documented.

In the following sections, we first review the literature on the role of play in the development of adult competency cross-culturally. Next, we outline our three theoretical proposals for the emergence of cooperative goal structures in games. We then model the data using phylogenetically-controlled multinomial, univariate and multivariate regressions. We conclude with a presentation of our findings and a discussion of the relevance of rule-based games to ongoing questions about cultural group selection and the evolution of human cooperativeness.

## Children's play

Play is a ubiquitous and essential activity that is believed to prepare children for their adult lives, both socially and in subsistence related tasks [1, 27]. Play allows children to imitate adults and helps them acquire culture-specific skills, norms, and behavioral repertoires [1]. Children learn emotional, physical, cognitive, and social skills during play [28], and play is vital for learning new strategies in unfamiliar situations [29]. Recent studies on children's activities in both small-scale foragers [15] and WEIRD [30] populations [31] suggest that children spend 25–30% of their time playing, making play a potentially substantial driver of the socio-cognitive development observed during childhood.

The specific kinds of play that children engage in, however, varies across cultural groups. For example, children of Aka foragers in the Central African Republic engage predominately in work-pretense play, whereas children of their Ngandu neighbors—living in a more socially stratified context—engage in more competitive forms of play [15]. Similar cross-cultural differences in children's play have been examined in other studies, contrasting, for example, German and Thai children [32], or children in Kenya, Mexico, the United States, India, Okinawa, and the Philippines [33]. Children also engage in different forms of play depending on age [32] and sex [15, 34] across cultural groups.

However, despite its omnipresence across diverse cultural contexts, play has been more thoroughly examined in Western, industrialized cultural groups than in small-scale cultural groups [1, 14, 15, 27, 35–37]. This sampling bias reduces the representativeness of the literature [38] and ignores substantial cross-cultural variation in children's playing behavior [39]. In this paper, we will attempt to build a more robust understanding of the potentially functional role that play serves cross-culturally, by investigating the variation in goal structures within a specific type of play known as *games*.

## Games

Games are a type of play characterized by predefined rules that structure interactions between players [40]. Game types (i.e., games of strategy, chance, physical skill, etc.) vary with geographic location [32], child-rearing practices [41], social complexity [14], and cultural norms [16, 17, 42]. As for the function of games, a study by Eifermann [17] suggests that Kibbutz children's games reflect cultural values of cooperation and egalitarianism. Ager's [16] study on the games played by a small Inuit community in Tununak, Alaska, mirrors these findings: the types of games and the resulting form of social interactions between players reflect Inuit cultural values of non-aggression and autonomy.

As with research on children's play more generally, extant work on games suffers from a sampling bias; wide-ranging comparative studies of the relationship between games and cultural characteristics are lacking. Rule-based games have even been explicitly excluded from previous research on the relationship between phylogeny and ontogeny of play [43], being regarded as inflexible and "not foster[ing] innovation" in children's behavior [29]. Moreover, to the extent that games have been studied cross-culturally, research has mainly focused on a single category of games—competitive ones [22]. For example, Roberts et al. [14] offered a

commonly used definition of games that numerous subsequent studies [41, 44–50] then adopted: *organized, competitive, and rule-based play between two or more players, with a winner.* This definition exemplifies the common understanding of games as competitive activities and has excluded other forms of games—such as cooperative or solitary ones—from the lenses of comparative research [22]. Due to this bias, little is known about how cultural selection pressures [8, 9] on the cooperativeness or competitiveness of games might emerge as a function of variation in the cultural and natural environment.

## Possible drivers of cooperative goal structures

Games can vary in their goal structures—some games emphasize cooperative behavior between individuals to achieve a shared goal (e.g., in hacky sack), while other games emphasize competitive behavior between individuals (e.g., in chess). Assuming that games help children acquire the skills needed to be successful as an adult in their local cultural context, and that games vary in the extent to which their goal structures emphasize cooperation, a question arises with respect to the cultural evolutionary drivers of the goal structures of games: which social and ecological forces might cause the goal structure of games to emphasize cooperation in one cultural group, and competition in another? As outlined earlier, we consider three possible explanations for the emergence and maintenance of cooperative goal structures: one based on the frequency of group foraging, one based on conflict prevalence and structuring, and the third based on social stratification.

**Interdependence in foraging.** Human groups [51–54], and even some non-human primates [55–57], rely on coordination and social learning in their food quests. If coordinated labor (or generalized interdependence [58] in subsistence) is an essential precondition of securing food [59], then we would predict that children who develop elevated competency in coordinated, goal-oriented actions with shared intentions (i.e., as found in cooperative games), would have higher foraging competence as adults. As we outlined earlier, if experience playing cooperative games over developmental time improves adult foraging competency in groups, then the goal structures of rule-based games should covary with the extent of interdependence in foraging.

This leads us to the prediction that:

**P1** The frequency of games with cooperative goal structures will be positively associated with the presence of interdependence in foraging.

**Intra- and inter-group conflict.** Bowles, Gintis, and Choi [10, 11, 60] argue that within-group cooperation can be stabilized by inter-group competition. They draw on a more general proof by Price [61] that selection between groups can overcome individual-level incentives to shirk cooperation, and provide a plausible causal mechanism linking two widely observed facts about human psychology: 1) humans are frequently altruistic (i.e., they often confer benefits on others at a cost to themselves), and 2) humans are frequently parochial (i.e., they often favor ethnic, religious, or group insiders over outsiders). Specifically, Bowles [10] argues that: "altruism would have facilitated the coordination of raiding and ambushing on a scale known in few other animals, while parochialism fuelled the antipathy towards outsiders." In unison, these forces can lead to strong selection pressures for intra-group cooperation, especially because there is a substantially greater scope for selection to act on cultural drivers of cooperative behavior than genetic ones [9, 62]. Zefferman and Mathew [63] elaborate on this argument more explicitly, arguing that: "human warfare meets the two necessary and sufficient conditions for group-structured cultural selection: Variation in cultural traits between groups

influences the outcome of warfare and the outcome of warfare influences the spread of these cultural traits."

Success in inter-group conflict depends on effective tactics, large-scale organization, and purposeful goal-oriented coordination between individuals. Again, as we have outlined before, if experience in game play under coordination- and cooperation-based goal structures over the period of childhood development improves adult competence in contexts demanding intra-group cooperation, then we would expect to see selection on the goal structure of games respond to the extent of inter-group warfare.

This leads us to the predictions that:

**P2a** The frequency of games with cooperative goal structures will be positively associated with the presence of inter-cultural warfare.
and:

**P2b** The frequency of games with cooperative goal structures will be negatively associated with the presence of intra-group conflict.

### Lack of social stratification

Social stratification refers to the unequal distribution and accessibility of valuable resources (e.g., education, income, power) within sub-groups of a given society [64]. Cross-cultural studies have found that social stratification varies with population size [65], human sacrifice [66], and belief in moralizing high gods [67]. Recent observational studies also suggest a relationship between social stratification and the types of play that children engage in [15, 16]. In one study, the socially stratified Ngandu were found to play more competitive games than their egalitarian Aka neighbors [15]. Two other investigations also suggest that egalitarian social structures are reflected in the structure of games [16, 17], in so far as egalitarian-structured cultural groups engage in more cooperative games than socially stratified cultural groups.

This leads us to the prediction that:

**P3** The frequency of games with cooperative goal structures will be negatively associated with the presence of social stratification.

## Methods

### Games

Our analysis draws on the goal structure codings provided by the AustroGames database [22, 68]. This open-access dataset provides detailed information on historical games played by cultural groups across the Austronesian language family. These data cover the ethnographic research period from the 18th to the 20th century.

We use the following game filtering steps in R [69] provided by Leisterer-Peoples, et al. [68]: i) games must have been linked to an Austronesian Basic Vocabulary Database code [24], ii) games must be described in enough detail to assign a goal structure code, iii) games must not be of non-local origin, iv) games must occur within cultural groups in the Austronesian language phylogeny [21], v) games must occur in cultural groups with covariate data in Pulotu [23], and vi) the game descriptions must correspond to the same time frame as the covariate data from Pulotu, ±50 years. Additional information on the sample sizes after each filtering

**Table 1. The distribution of goal structures in the original dataset (after filtering steps)** [22] **and in the distribution used in the analyses.** The data were grouped into three categories for the analyses in the current study: cooperative, competitive, and solitary (see Fig 1 for more details on the goal structure types and the S1 File for the number of games in each of the 25 ethnolinguistic groups).

| Goal structure | Original sample size | Analysis sample size |
|---|---|---|
| Solitary | 23 | **23** |
| Competitive | 76 | 85 |
| Competitive vs. Solitary | 9 | |
| Competitive vs. Cooperative group | 8 | **60** |
| Cooperative group vs. Cooperative group | 45 | |
| Cooperative | 7 | |
| **Total** | 168 | **168** |

step, and the biases that could potentially be introduced by these filters, can be found in the S1 File.

Due to small sample sizes in several of the goal structure categories (e.g., purely cooperative games; see Table 1 for sample sizes and Fig 1 for details on the goal structures), we collapse the goal structures of games into three main groups: *cooperative games* (i.e., all games with cooperative interactions: cooperative group, cooperative group versus cooperative group, competitive versus cooperative group), *competitive games* (i.e., games without cooperative interactions, but with competitive interactions: competitive versus solitary, competitive), and *solitary games* (i.e., games with neither cooperative nor competitive interactions: solitary).

## Cultural covariate data

Socio-cultural and ecological data from cultural groups in the Austronesian language family were obtained from the Pulotu database [23]. The Pulotu database documents both historical and modern-day cultural and religious data from 116 geographically diverse ethnolinguistic groups in the Pacific [23].

**Interdependence in foraging.** The cultural groups included in our analyses are mainly located within the Pacific region; as such, both land- and water-based hunting are vital sources of sustenance for many of the cultural groups. Subsistence activities can vary in the amount of coordination and cooperation they require [70]. We therefore assess the presence of substantial group-based hunting on land and water.

Land-based group hunting, and water-based hunting and fishing in groups [23], were independently coded as present or absent in each population. Each group-based hunting style was coded as *present* if a substantial portion of the diet was produced under such a hunting style. A given group-based hunting style is considered *absent* if it is either not practiced or if it does not contribute substantially to the typical diet.

For example, the people of Buka, an island in Bougainville, eastern Papua New Guinea, acquire a large portion of their sustenance through water-based hunting in groups; therefore, water-based hunting is considered *present* among the people of Buka. The people of Buka also hunt pigs for special occasions. For example, it is stated that: "Pig being a luxury usually reserved for feasts, and other flesh food being somewhat scarce and hard to come by, the usual relish to a meal of taro consists of fish." [71]. Because a substantial portion of the diet on Buka does not result from land-based hunting in groups, land-based hunting in groups is considered *absent* there.

**Conflict.** The frequency and intensity of conflict and warfare is measured in three variables in Pulotu [23]: i) the frequency of inter-personal conflict *within* local communities of a

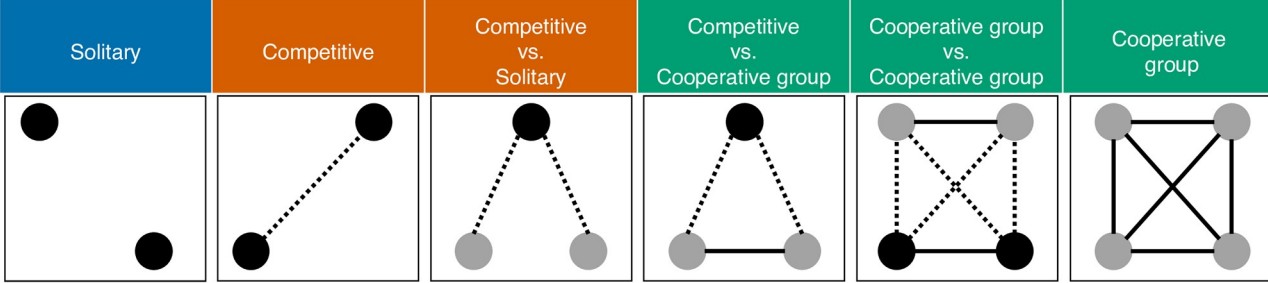

**Fig 1. The goal structure coding of games from Leisterer-Peoples et al. [22] (reproduced under CC-BY 4.0 license).** Each dot represents a player. The color of the dot represents the goal of the player; players with differing goals have different colored dots. A dotted line represents competition and a solid line cooperation. The absence of a line represents a lack of either cooperation or competition between players. The number of dots and lines is simplified for illustration (i.e., each goal structure can have more than 2–4 players without necessarily changing the structure of the game). For example, tag is a game with a "competitive vs. solitary" goal structure—a lone competitive player (i.e., the black dot) tries to tap the other players (i.e., the gray dots), who each have the same goal of staying untagged, but are neither cooperating nor competing with one another. The goal structures are grouped into three main categories for the current study: games with cooperation (green label), competitive games (red label), and solitary games (blue label).

given ethnolinguistic group, ii) the frequency of conflict *between* local communities of the same ethnolinguistic group, and iii) the frequency of conflict with *other* ethnolinguistic groups. These variables were re-coded for the current study as follows:

Intra-group conflict (i.e., between individuals or small groups within the same local community), was considered *present* if such conflict was frequent and often violent, and *absent* if such conflict was either rare, or frequent but seldom violent.

Intra-cultural conflict (i.e., conflict occurring between two or more villages or sub-populations of the same ethnolinguistic group), was considered *present* if such conflict occurred more often than once in a generation, and *absent* if such conflict took place roughly every generation or never occurred.

Inter-cultural warfare, or conflict with other ethnolinguistic groups, was considered *present* if such conflict occurred more often than once in a generation, and *absent* if such conflict took place roughly every generation or never occurred.

**Social stratification.** The data on social stratification were provided by Watts et al. [25]. Social stratification was considered *present* if there were intergenerational differences in wealth, and/or if status and mobility between social positions was restricted. Social stratification was considered *absent* if ethnolinguistic groups were egalitarian or if individuals could feasibly change their wealth, status, or social position.

**The Austronesian language phylogeny.** The Austronesian language phylogeny used in this study was reconstructed by Gray et al. [21] using 210 basic vocabulary items. The ethnolinguistic groups corresponding to each game were matched to the primary indigenous language of the region in the Austronesian Basic Vocabulary Database [24] and the Pulotu database [23]. In many cases, there was no match (i.e., the ethnolinguistic group did not correspond to any branch on the language phylogeny) and those games were excluded from further filtering and analyses. The Austronesian language phylogeny was subsequently pruned to the 25 ethnolinguistic groups included in the analyses.

## Statistical analyses

We use univariate and multivariate, multinomial regressions using a Bayesian framework, coded in Stan [72], with and without phylogenetic controls, to estimate the relationship between the frequency of game goal structures and the presence or absence of cultural variables. In these models, the outcome variable is the count of games in each of the goal structure

categories within each ethnolinguistic group. The predictors are binary variables indicating the presence or absence of each cultural variant within each ethnolinguistic group. Full model descriptions are included in the S1 File. We use weakly regularizing priors to prevent overfitting of the sample, but we do not apply corrections with regard to multiple testing, as we are using a Bayesian analysis framework, as opposed to a null-hypothesis rejection framework [73]. The data used in the analyses, along with our statistical and pre-processing code, are available on GitHub: https://github.com/sarahpeoples/AustroGamesGoalStructures and Zenodo [74].

Models without controls based on the Austronesian language phylogeny assume that each ethnolinguistic group can be treated as an independent data point for the purpose of model fitting. However, these groups arguably share linguistic and cultural ancestry [21] that may have introduced correlations in outcomes. This shared ancestry is caused by the diversification of cultural groups over time whereby daughter cultures inherit many of the traits of their parent cultures before subsequently diverging themselves. Therefore, we use the Austronesian language phylogeny as a proxy for underlying cultural history [66, 75–79]. Our models use the language phylogeny to introduce correlated random effects, which help to address Galton's Problem [80, 81] and account for the potential non-independence of the ethnolinguistic groups in our study.

## Results

### Descriptive statistics

A total of 168 games from 25 ethnolinguistic groups were included in the analyses. Purely competitive games were the most common type of game across ethnolinguistic groups ($n = 76$), followed by cooperative group versus cooperative group games ($n = 45$) and purely solitary games ($n = 23$). The frequency of each goal structure types in our study is given in Table 1, and the distribution of cultural variables is visualized in Fig 2. In 14 of the 25 ethnolinguistic groups, at least half of the games had competitive goal structures. In 7 of the 25 ethnolinguistic groups, at least half of the games had cooperative goal structures. For 11 of the ethnolinguistic groups, the outcome vector consisted of only 1 to 3 games. More detailed information about the distribution of goal structures across ethnolinguistic groups in our study is included in the S1 File.

### Cultural variables and goal structures

**Univariate models.** Our univariate regression analyses indicate associations between the goal structure of games and both subsistence and conflict measures. Specifically, we find that, as predicted in **P1**, interdependence in land-based subsistence is associated with increased log-odds of cooperative goal structures ($\beta = 0.92$, 90% CI [0.30, 1.51]), but contrary to our prediction, interdependence in water-based subsistence is not ($\beta = -0.24$, 90% CI [−0.83, 0.32]). Likewise, as predicted in **P2a**, frequent war or conflict with other cultural groups is associated with increased log-odds of games with cooperative goal structures ($\beta = 0.81$, 90% CI [0.22, 1.37]). And as predicted in **P2b**, frequent intra-group conflict is associated with decreased log-odds of games with cooperative goal structures ($\beta = -0.86$, 90% CI [−1.47, −0.32]). We find no support for **P3**, as there is no association between social stratification and the log-odds of games with cooperative goal structures ($\beta = 0.17$, 90% CI [−0.41, 0.75]). See Fig 3 for posterior densities covering each of these cases.

**Multivariate models.** Given the observed associations between both subsistence and conflict variables and our outcome of interest, and the potential causal connections between these predictor variables, we also fit multivariate models including both predictors. Table 2 indicates

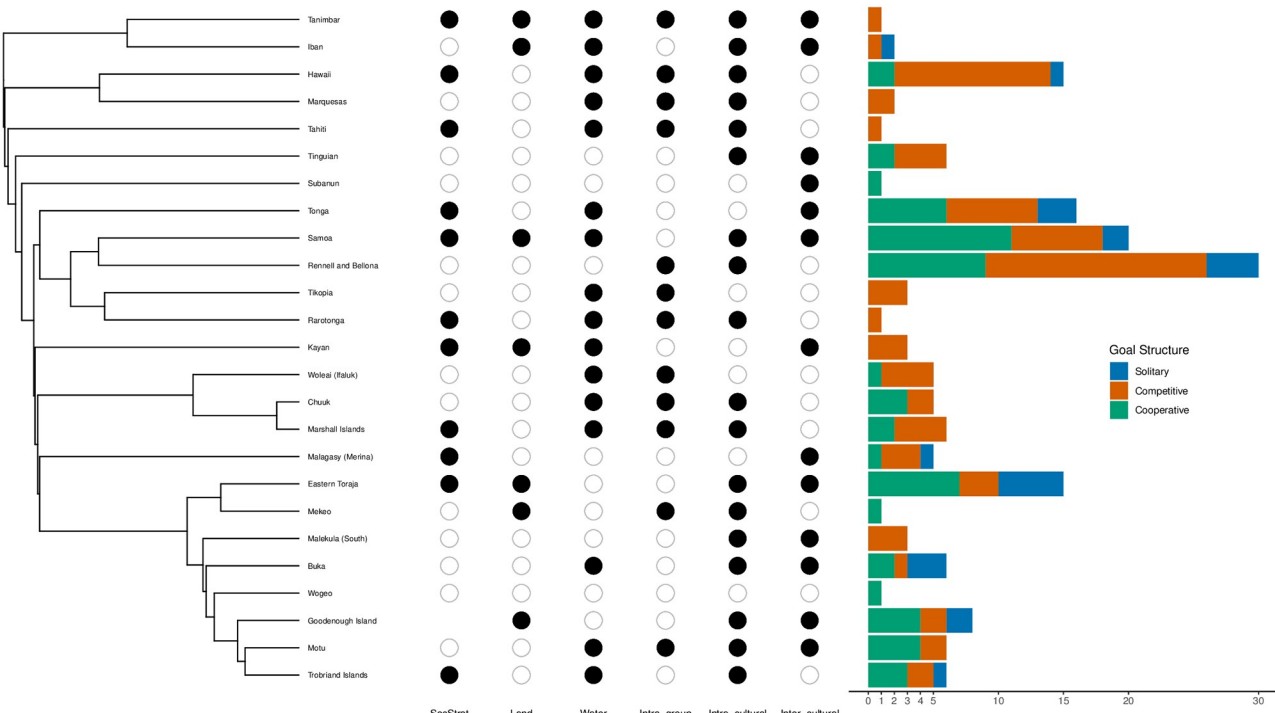

**Fig 2. The distribution of game goal structures and associated cultural variables.** The distribution of game goal structures and associated cultural variables mapped onto the pruned Austronesian language phylogeny [21]. The dots represent the presence (i.e., the black dot), absence (i.e., the white dot), or missing data (i.e., no dot) of each cultural variant (i.e., social stratification, interdependence in land-based hunting, interdependence in water-based hunting and fishing, intra-group conflict, intra-cultural conflict, and inter-cultural conflict). The bar graphs represent the frequency distribution of game goal structures by cultural group (i.e., blue represents solitary games, orange represents competitive games, and green represents cooperative games). The sample sizes of games in each cultural group vary as shown. More details can be found in the S1 File.

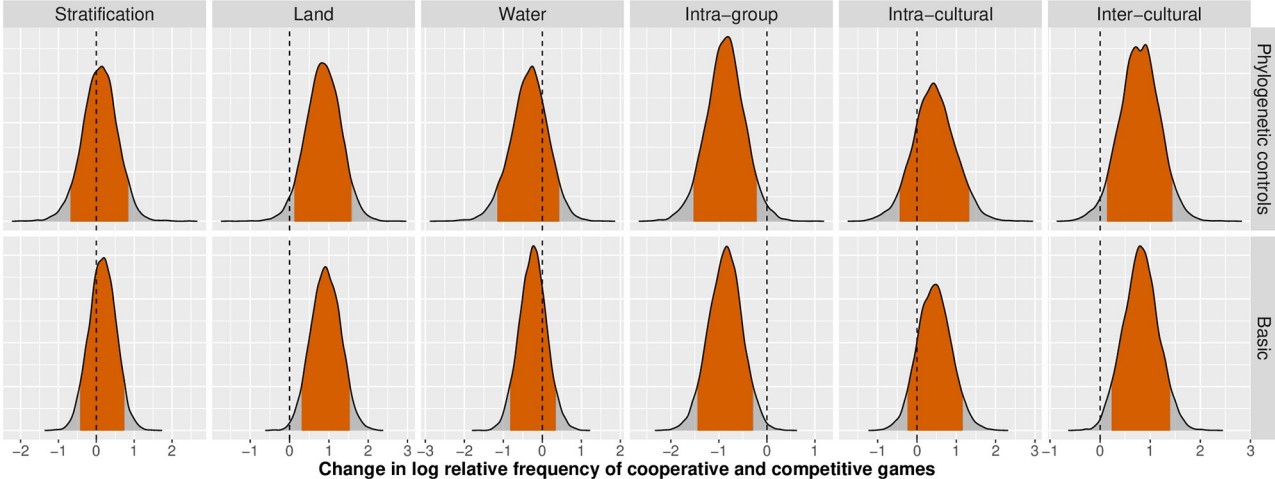

**Fig 3. Univariate model results.** The change in log-odds of cooperative relative to competitive games as a function of predictor variables, with and without phylogenetic controls included in the model. A positive parameter value corresponds to an increase in the relative frequency of cooperative games to competitive games, while a negative parameter value corresponds to the opposite effect. The shaded areas indicate the 90%-credible intervals. We note positive effects of interdependence in land-based hunting and inter-cultural conflict as predictors of cooperative goal structures in games. Frequent intra-group conflict is associated with a lower frequency of cooperative games. Additional plots are included in the S1 File.

**Table 2. Regression results.** For each model (*Model*), we present the effect of a given predictor variable on the log-odds of cooperative versus competitive games—positive values indicate variables that are associated with an increased relative frequency of cooperative goal structures. The top block of estimates are from models omitting phylogenetic controls (*Phylo.*), and the bottom block of estimates are from models that control for phylogenetic signal in the outcome using phylogenetically constrained random effects. The top block corresponds with the "Basic" plots in Fig 3 and the bottom block with the "Phylogenetic controls" plots in Fig 3. In the univariate models, the column *Predictor* gives the effect of the indicated variable. In the multivariate models, the column *Predictor* gives the effect of the indicated variable and the column *Interdependence land* gives the effect of interdependence in land-based hunting. Intervals are central 90% posterior credible intervals. *D* gives the relative WAIC (Watanabe-Akaike Information Criterion) value compared to the best model, and *W* gives the unit normalized WAIC weight across both phylogenetic and non-phylogenetic models.

| Model | Phylo. | Predictor | Interdependence land | D | W |
|---|---|---|---|---|---|
| Base model | | | | 10.018 | 0.002 |
| Interdependence land | | **0.92 (0.30; 1.51)** | | 5.375 | 0.02 |
| Interdependence water | | -0.24 (-0.83; 0.32) | | 13.825 | 0 |
| Intra-group conflict | | **-0.86 (-1.47; -0.32)** | | 1.662 | 0.13 |
| Intra-cultural conflict | | 0.44 (-0.25; 1.14) | | 13.577 | 0 |
| Inter-cultural conflict | | **0.81 (0.22; 1.37)** | | 4.157 | 0.037 |
| Social stratification | | 0.17 (-0.41; 0.75) | | 14.71 | 0 |
| Intra-group conflict | | -0.59 (-1.282; 0.036) | 0.57 (-0.19; 1.27) | 3.908 | 0.042 |
| Inter-cultural conflict | | 0.50 (-0.19; 1.21) | 0.62 (-0.16; 1.35) | 6.354 | 0.012 |
| Base model | ✓ | | | 1.968 | 0.111 |
| Interdependence land | ✓ | **0.86 (0.12; 1.56)** | | 3.665 | 0.048 |
| Interdependence water | ✓ | -0.32 (-1.15; 0.42) | | 3.604 | 0.049 |
| Intra-group conflict | ✓ | **-0.86 (-1.51; -0.20)** | | 0 | 0.298 |
| Intra-cultural conflict | ✓ | 0.42 (-0.41; 1.34) | | 5.948 | 0.015 |
| Inter-cultural conflict | ✓ | **0.79 (0.14; 1.43)** | | 2.08 | 0.105 |
| Social stratification | ✓ | 0.10 (-0.62; 0.88) | | 6.14 | 0.014 |
| Intra-group conflict | ✓ | -0.62 (-1.35; 0.15) | 0.53 (-0.35; 1.30) | 2.511 | 0.085 |
| Inter-cultural conflict | ✓ | 0.51 (-0.29; 1.23) | 0.57 (-0.25; 1.42) | 4.645 | 0.029 |

that accounting for conflict variables substantially reduces the effect of interdependence in land-based subsistence on the log-odds of cooperative goal structures from $\beta = 0.92$ (90% CI [0.30, 1.51]) in the univariate model to $\beta = 0.57$ (90% CI [−0.19, 1.27]) when intra-group conflict is included in the model, and to $\beta = 0.62$ (90% CI [−0.16, 1.35]) when inter-cultural conflict is included in the model; see column labeled *Predictor 2* in Table 2. Similarly, the association between intra-group conflict and the goal structure of games reduces when accounting for the association between interdependence in land-based subsistence and goal structure in the model, $\beta = −0.59$ (90% CI [−1.282, 0.036]). A similar pattern is found with the association between inter-cultural conflict and the goal structure of games, $\beta = 0.50$ (90% CI [−0.19, 1.21]).

Additionally, when models include phylogenetic controls, we find support for the association between interdependence in land-based subsistence and the goal structure of games, $\beta = 0.86$ (90% CI [0.12, 1.56]). The models with phylogenetic controls continue to show support for a positive association between frequent conflict with other cultural groups and the log-odds of games with cooperative goal structures, $\beta = 0.79$ (90% CI [0.14, 1.43]) and a negative association between frequent intra-group conflict within a given cultural group and the log-odds of cooperative games, $\beta = −0.86$ (90% CI [−1.51, −0.20]). These patterns are reduced when interdependence in land-based subsistence is included in models with intra-group conflict and inter-cultural conflict (intra-group conflict: $\beta = −0.62$, 90% CI [−1.35, 0.15]; inter-cultural conflict: $\beta = 0.51$, 90% CI [−0.29, 1.23]). Similarly, the effect of land-based subsistence is reduced to $\beta = 0.53$ (90% CI [−0.35, 1.30]) when intra-group conflict is included in the model, and to $\beta = 0.57$ (90% CI [−0.25, 1.42]) when inter-cultural conflict is included in the model.

## Discussion

The current study has investigated associations between group-level measures of interdependence in subsistence, conflict, social stratification, and the goal structure of games. We find evidence of a moderately robust association between the cooperativeness of game goal structures and (i) measures of conflict and (ii) a measure of interdependence in subsistence. In cultural groups with frequent inter-cultural warfare, games tend to be cooperatively structured—an effect driven in large part by games that feature cooperative groups competing with other cooperative groups (these kinds of games make up 72% of the cooperative games in our sample, see Table 1). Similarly, competitive games are played at higher rates in cultural groups with higher levels of intra-group conflict. We also found support for a positive relationship between elevated interdependence in subsistence and the frequency of cooperative games. Our findings are consistent with theories of cultural group selection, which link competition between groups to the evolution of cooperative institutions within groups [9, 10, 63]. Though they are an often overlooked cultural institution, we argue that games may play an important role in the acquisition of locally-relevant adult competence by children and teenagers.

Proponents of cultural group selection [9, 63] have proposed that there are two conditions needed for warfare to cause selection at a group-level: 1) variation in cultural traits must influence the outcome of warfare, and 2) the outcome of warfare must influence the spread of these traits. Our results provide direct support for neither of these two conditions; rather, they establish a simple association between the existence of inter-cultural warfare and the more frequent practice of games that feature cooperative groups competing with other cooperative groups. This positive association, however, is to be expected under the cultural group selection model [9, 63]. A causal relationship is plausible for a few reasons: 1a) small-scale warfare is a complex, high-stakes situation, and the success of a given group in such a situation is likely dependent on adaptations that enforce within-group organization and coordination [82–85], and 1b) years of experience playing games with cooperative goal structures over developmental time should help individuals learn to organize behavior around shared goals and deploy coordinated actions, yielding at least some marginal advantage relative to groups with less experience in organizing into cooperative units. Additionally, 2) if groups with cooperative norms do outcompete other groups in direct conflict, the local frequency of such cooperative norms should increase, as territory and resources are generally seized by the victors of inter-group conflicts [86]. As such, we suspect that the association of inter-cultural and intra-group conflict with the goal structure of games may be the result of a causal selection process, as appears to be the case with many cultural norms [79]. However, our study is only an exploratory first step toward making such a case and further details will need to be validated using additional datasets and methods.

Extant ethnographic research on play lends some support to our arguments. Boyette [15], for example, finds that play in a modern-day foraging cultural group and a modern-day pastoralist cultural group, occupies the majority of children's daily lives. The frequency with which children engage in play highlights the potential importance of play and games in the transmission of cultural norms. Likewise, some experimental research examining the effect of games on German children's prosocial behavior provides evidence that playing cooperative games increases children's willingness to share with third-parties [87]. Experiments investigating Western children's understanding of rules show that 5-year-old children understand and enforce norms of cooperation in a game-like setting [88]. In sum, these findings suggest that experience with games that have cooperative goal structures over developmental time may have effects on real-world behavior in more general contexts.

It is important to note that the reliability of the effects of our conflict variables was reduced when land-based hunting in groups was included as a covariates in the models. As such, the goal structure of games does not appear to be robustly linked to conflict in our small dataset. Future studies would benefit from a larger sample of cultural groups, and more precise covariate measures.

Future studies evaluating the plausibility of our arguments might additionally study: 1) if the behavioral, organizational, and strategic practices employed in games mirror generally parallel practices in situations of actual conflict [89, 90], 2) if individuals or groups with greater experience playing cooperative games are empirically more likely to be victorious in real-world competitive settings [91–93], and 3) if there is evidence of games with cooperative goal structures actually spreading as a consequence of military conflict [94]. Additionally, future studies might investigate the effects of preference for game goal structures at an individual level, as some have criticized the focus of CGS on group-level variation [95, 96]. Finally, future studies might also investigate whether gender-division plays a role in the distribution of games within cultures. Given that our cultural covariates are activities primarily performed by males, it might be that variation in the sex of game players across cultures has influenced the relationships uncovered here. Unfortunately, the AustroGames database does not currently offer information on the sex of the players as this is under-described in the primary literature. Future studies should focus on gathering further ethnographic materials that mention such details.

Subsistence activities are often a focus of play in early childhood [97]—children imitate adult behavior in work pretense play, such as nut-cracking, gathering berries, or cooking a meal, before they substantially contribute to sustenance acquisition. Such play is specifically thought to help children prepare socially and behaviorally for life as successful adults in a given socio-cultural context [1]. Subsistence style has also been shown to vary with child training [98] and individual conformity [99]. In line with one of our predictions, games with cooperative goal structures are more frequent than competitive games when interdependence in land-based subsistence is present. However, overall, our analyses lend only moderate support for a linkage between subsistence mode (as measured by the specific Pulotu variables used in our models) and the cooperativeness of game goal structures.

First, the association between interdependence in subsistence and the goal structure of games was not replicated in both land and water-based subsistence measures. Given the importance of water-based subsistence in the Pacific, this failure to replicate is important to emphasize. Second, the effect of interdependence in land-based hunting was not reliably positive after including intra-group conflict or inter-cultural warfare as co-variates in the models. Additionally, the interdependence in land-based subsistence models (with and without phylogeny) received little weight in the WAIC comparison in Table 2, indicating weaker support for this association. As such, the goal structure of games does not appear robustly linked [79] to interdependence in foraging in our small dataset.

Differences in the levels of cooperative behavior needed for success in specific subsistence styles have been suggested in the literature. For example, Talhelm et al. [70] suggest that differences in the cooperative action required to farm wheat versus rice crops affect group behavior. We used open-access data from Pulotu [23] as a proxy for real-world interdependence in subsistence. It is possible that these simple, group-level binary variables are weak or noisy measures of the importance of sustenance-oriented interdependence behavior. Future studies should aim at defining real-world measures of group-level cooperation in subsistence and in other activities. Given the historic nature of the game data, we were unable to use economic games to measure levels of cooperation among groups of individuals engaging in specific tasks

[100–102], but experimental methods have proven useful in measuring similar constructs in other places [103].

We do not find evidence of a relationship between the level of social stratification in a given cultural group and the goal structure of games. Previous studies indicating a relationship between social stratification and the cooperativeness of games [15–17] do so descriptively. However, by drawing on a larger set of comparative data, our approach has more power to study the distribution of the goal structure of games as a function of group-level variables. While our study suggests that the distribution of goal structures in games across cultures might not depend on social stratification, we note that our group-level stratification variable is a simple, reductive measure. Future research should explore possible relationships between social stratification and the goal structure of games using more informative variables on the within-group structuring of inequality. Previous studies have argued that causal linkages between group-level variables and social/material inequality can depend not only on the overall level of inequality within a society, but on how such inequality is structured [104].

The current study investigates the relationship between the goal structure of games and associated socio-ecological variables among 25 ethnolinguistic groups in the Austronesian language family. We have focused our initial data collection and coding efforts [22] on these groups because of the large cultural and linguistic diversity expressed in this area of the world [21] and the availability of previously published socio-cultural data [23] that could be linked to the game data [22]. Nevertheless, only 168 games from 25 ethnolinguistic groups were described in enough detail to warrant inclusion into our analyses—leaving our sample fairly small and sensitive to possible false positives (and negatives). In light of recent focus on replication in science, future studies should examine if the goal structures of games played in other parts of the world, including non-Western cultures, show a similar patterning with respect to conflict variables. Such studies would help to evaluate the robustness of the findings presented here across a wider sample of cultural groups. Additionally, future studies could examine the relationship between games played in contemporary cultural groups and group-level cultural variables.

A key limitation of our study beyond sample size is that we model the *count* of games by goal structure. Theoretically, the *amount of time* spent engaging in games of a given goal structure would be a more precise measure of a player's experience with that goal structure. Such fine-scale time allocation data across games and cultural groups, however, does not yet exist. Also, the AustroGames dataset [22] does not currently provide information on the typical ages of players of each game. Investigations of adult engagement in traditional games are limited, but interviews in one Inuit community suggest that games may play a role in young adults feeling "connected" with their cultural group [105].

The cultural variables considered in this study were pragmatically chosen based on the availability of corresponding open-access data, and are by no means exclusive. Future studies should investigate additional (theory-driven) explanations for the cooperativeness of games and their distribution throughout the world: though we believe that subsistence and conflict variables capture important group-level characteristic that might influence the evolution of specific cultural practices, other causal paths may be as or more important.

An additional limitation of the current study is the assumption of objectivity and longitudinal stability in the ethnographic record [106]. The Pulotu and AustroGames datasets are based on published ethnographic and historical accounts, which are reliant on the neutrality [107] of the ethnographer and the time period during which the ethnographer visited the cultural group. By focusing on a different time period or on different ethnographic accounts, the data and the effects found here could also be subject to change. Future studies should investigate the longitudinal stability [107] of cultural variables (e.g., the extent of conflict between cultural

groups) and should explore the relationship between games and cultural variables as cultures adapt to external circumstances (e.g., missionaries, colonists, globalization). For example, how robust is the relationship between the goal structure of games and levels of conflict over time? Do games rapidly change with changes in conflict frequency (or is there a time lag)?

Due to sample size constraints, we grouped the six original goal structure categories of games into three broad categories—solitary, competitive, and cooperative (see Fig 1). Cooperative group versus cooperative group games were categorized as cooperative games. However, cooperative group versus cooperative group games could also be seen as an independent kind of goal structure. Cooperatively-structured competition is important in human psychology [108] and cultural group dynamics [60], and should therefore be thoroughly examined in future investigations with larger samples.

Aside from cooperation and competition, the current study did not examine the particular skills that are transmitted through games. This leaves the proximate question as to which specific physical, cognitive, emotional, and social skills players learn through games that might increase their success in real-world cooperative dilemmas. Future studies should examine how other aspects of games, aside from their goal structures, might influence cooperative and competitive actions. Games do not only transmit cultural norms to players, but also allow players to practice and acquire skills that can be useful in warfare or subsistence (e.g., hand-eye coordination, event planning, emotional stability, and strategic thinking). Early cross-cultural examinations of the relationship between the type of games (i.e., games of strategy, physical skill, and chance) and cultural variables suggest that games of strategy frequently imitate war activities [14, 109]. Future studies should re-examine and elaborate on these findings using modern statistical and methodological tools, such as phylogenetic comparative methods [79].

## Conclusion

Non-human animals engage in play(ful) behavior [110–115], but only humans play rule-based games [112]. Rule-based games are more than just child's play. Our study provides evidence for a relationship between the goal structure of games and the social and ecological environments of the ethnolinguistic groups that play them. The type and intensity of conflict, as well as the extent of interdependence in the acquisition of sustenance on land, appear to be correlated with the occurrence of specific types of game goal structures in our dataset. This evidence, though correlational, contributes to a growing body of literature which suggests that games may play a functional role in human culture by mimicking overt real-world behavior, thus serving as training grounds for norms and behaviors that are relevant for a particular socio-ecological context. However, future experimental studies with larger sample sizes would be needed to verify such predictions.

## Supporting information

**S1 File. Additional information on analyses and results.**
(PDF)

## Acknowledgments

Thanks to Heidi Colleran, Cara Evans, Joseph Watts, and Roman Stengelin for helpful feedback on earlier versions of this manuscript.

## Author Contributions

**Conceptualization:** Sarah M. Leisterer-Peoples, Cody T. Ross, Susanne Hardecker, Daniel B. M. Haun.

**Data curation:** Sarah M. Leisterer-Peoples, Simon J. Greenhill.

**Formal analysis:** Sarah M. Leisterer-Peoples, Cody T. Ross.

**Methodology:** Sarah M. Leisterer-Peoples.

**Project administration:** Sarah M. Leisterer-Peoples.

**Software:** Sarah M. Leisterer-Peoples, Cody T. Ross.

**Supervision:** Susanne Hardecker, Daniel B. M. Haun.

**Visualization:** Sarah M. Leisterer-Peoples, Cody T. Ross.

**Writing – original draft:** Sarah M. Leisterer-Peoples, Cody T. Ross.

**Writing – review & editing:** Sarah M. Leisterer-Peoples, Cody T. Ross, Simon J. Greenhill, Susanne Hardecker, Daniel B. M. Haun.

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
