## [Decision Letter · Decision Letter 0]

1 Sep 2021

PONE-D-21-23199

Games and enculturation: A cross-cultural analysis of games and values in Austronesia

PLOS ONE

Dear Dr. Leisterer-Peoples,

Thank you for submitting your manuscript to PLOS ONE. After careful consideration, we feel that it has merit but does not fully meet PLOS ONE’s publication criteria as it currently stands. Therefore, we invite you to submit a revised version of the manuscript that addresses the points raised during the review process.

You will find two reports (PDF and pasted below). Both reports are fairly positive. Referee #2 is making a number of suggestions in order to clarify certain aspect of the paper. Referee #1 is much shorter. As both reviewers state this paper is interesting for a broad audience and therefore you need to clarify certain details in order to nail it for a more general audience. For instance, both of them indicate that they are not familiar with certain parts of the methodology. In this regard, Referee #2 provides a very extensive guide in order to put clarify certain parts of the ms. All in all, their reports are positive.

We look forward to receiving your revised manuscript.

Kind regards,

Pablo Brañas-Garza, PhD Economics

Academic Editor

PLOS ONE

“This project was funded by the Max Planck Society for the Advancement of Science.”

We note that you have provided information within the Acknowledgements Section. Please note that funding information should not appear in the Acknowledgments section or other areas of your manuscript. We will only publish funding information present in the Funding Statement section of the online submission form.

“This study was funded by the Max Planck Society (www.mpg.de). The funders had no role in study design, data collection and analysis, decision to publish, or preparation of the manuscript.”

Reviewers' comments:

Reviewer's Responses to Questions

**Comments to the Author**

1. Is the manuscript technically sound, and do the data support the conclusions?

Reviewer #1: Yes

Reviewer #2: Partly

2. Has the statistical analysis been performed appropriately and rigorously? 

Reviewer #1: I Don't Know

Reviewer #2: Yes

3. Have the authors made all data underlying the findings in their manuscript fully available?

Reviewer #1: Yes

Reviewer #2: Yes

4. Is the manuscript presented in an intelligible fashion and written in standard English?

Reviewer #1: Yes

Reviewer #2: No

5. Review Comments to the Author

Reviewer #1: ************************************************************************************

My comments are minor and they are available in the attached file. ***************

Reviewer #2: General points.

1) I should state that I am reviewing this manuscript from the perspective of a behavioral and experimental economist who has an interest in experiments using games. I will not be using the framework that a cultural anthropologist would bring to the paper, and so the authors and the editor should adjust their evaluation of my review accordingly.

2) I will make reference to some specific elements of the published criteria for PLOS-ONE acceptance as well as the PONE Reviewer Guidelines .

Specific Comments

3) PONE reviewer guidelines ask “What are the main claims of the paper and how significant are they for the discipline?” This paper claims to show evidence that the distribution of games played within a family of small-scale societies is not random. They test for relationships among several factors potentially related to cultural evolutionary selection pressures, and find some evidence that three factors: land-based foraging interdependence, intra-group conflict, and inter-cultural conflict, are predictive of the number of cooperative and/or competitive games. Specifically, intra-cultural conflict and land foraging interdependence are associated with more cooperative games, while intra-group conflict is associated with more competitive games. Speaking from the perspective of 1) above, this is significant. (I also think it is quite interesting, but that is not a review criterion.) The field of anthropologically-informed cross-cultural comparisons has been undergoing a renaissance in recent years, and this paper is a contribution to that stream of research. Overall, if the authors can address the other concerns I have raised, I believe this will be a publishable MS.

4) PONE criteria for publication state “Results reported have not been published elsewhere.” The material submitted constitutes a new analysis, and is a natural progression from the earlier publication by a number of the same authors of a description of the taxonomy of games in this family of cultures.

5) PONE reviewer guidelines ask “Is the manuscript well organized and written clearly enough to be accessible to non-specialists?” Here I have a couple of concerns that I suggest should be addressed before the MS is acceptable for publication. I start with two minor issues and progress to a more serious one.

a. From the standpoint of (1) above, a minor point is that a technical term used in a manner specific to cultural anthropology is nowhere defined: “phylogenetic”. (This could go in a footnote or the appendix, if the authors view it as too annoying to anthropological readers to put in the main text.)

b. The authors did not follow the submission guidelines for tables, which state “Tables are inserted immediately after the first paragraph in which they are cited.” They put Table 1 of the main text in the Supplementary Materials, and completely omitted the key main text results table, Table 2. I was able to get a version of Table 2 by request to the editor, and what I got was not in publication-ready format, but an excel spreadsheet. So, first of all, the authors need to get the correct publication-formatted tables in the right place. But second, I was quite puzzled to see that Table 2 is in a format with each model in a row. Now I may be at cross purposes with the way anthropologists regularly do things, but this does not make sense to me. I suggest that the authors re-format so that each model is a column (this is certainly the standard for anything remotely econometric in flavor). This is much easier to understand, in my opinion. (Of course, if I really am at cross purposes to the regression table norm for anthropology, the authors are free to ignore this suggestion.)

c. There is some kind of textual confusion in the MS, as a result of which it is just not clearly specified how the identified games are grouped for the statistical analysis. The statement about categories starting at line 196 on page 8/13 (starting with “Due to small sample sizes . . .”) presumably was intended to spell this out. However, there is a period in the middle of what was perhaps intended to be a sentence, and the verb or verbs are missing. So, the authors just do not clearly explain themselves on this point. This needs to be repaired.

6) I have a final set of concerns that flow from PONE guidelines and standards.

a. PONE Reviewer Guidelines state “Conclusions are presented in an appropriate fashion and are supported by the data.” On this point the authors generally do a good job in the main text. However, they relapse back to statements that are a bit too unqualified in the conclusion. The evidence presented is suggestive, not conclusive. And it is consistent with the view that “games can play a functional role in human culture by mimicking overt real-world behavior, serving as cultural training grounds”, but while this is currently our best conjecture at the relationship, it is not established by this paper. The abstract is considerably better in its wording.

b. The PONE submission guidelines state: "Include details of any corrections applied to account for multiple comparisons. If corrections were not applied, include a justification for not doing so". I believe I am correct in saying the authors have missed the ball on this one. I should be clear that in this kind of exploratory research using unique cultural/historical data, it is not feasible to collect new samples from the same population, and so multiple comparisons are appropriate to explore the patterns in the data. The authors need to acknowledge, however, that the statistical significance levels have to be taken with a certain grain of salt, because of data limitations.

c. The authors mention the sample size and potential bias issues that flow from the limitations of their data (page 18, line 409 and following lines on page 19.) The selection process is described starting on page 8, line 184, and following lines on page 9. What is missing are the statistics needed to see how the pruning of the initially broad list of games and societies was undertaken, along with any more specific comments about the potential for bias. For instance, in what order were criteria applied, and how many potential cases were lost due to each criterion? Additionally, I would expect to see some kind of statement as to what can be said about dropped cases, to the extent this is possible. For instance, do they vary from included cases in terms of any observable variables? I will speculate that not a lot can be said, and if so, perhaps this account will make up a paragraph or two in the supplemental materials that is referred to in the main text, but in my view something needs to be added. This is an exploratory work examining the relationship of multiple potential evolutionarily relevant causal factors to a specific type of outcome variable, but the lessons from narrower policy evaluation work are still relevant: it is always important to look out for selection effects that may be confounded with treatment effects. With the current paper’s data there will be limited ability to do this, and the authors have noted the issue, but I suggest little more attention should be paid to what can be said about this issue from the data.

6. PLOS authors have the option to publish the peer review history of their article (what does this mean?). If published, this will include your full peer review and any attached files.

Reviewer #1: No

Reviewer #2: No

---

## [Author Response · Author response to Decision Letter 0]

23 Sep 2021

Please see the "Response to Reviewers" PDF. I have provided detailed responses to both the reviewer and editor comments in that document. Thank you.

---

## [Decision Letter · Decision Letter 1]

26 Oct 2021

Games and enculturation: A cross-cultural analysis of cooperative goal structures in Austronesian games

PONE-D-21-23199R1

Dear Dr. Leisterer-Peoples,

We’re pleased to inform you that your manuscript has been judged scientifically suitable for publication and will be formally accepted for publication once it meets all outstanding technical requirements.

Kind regards,

Pablo Brañas-Garza, PhD Economics

Academic Editor

PLOS ONE

Additional Editor Comments (optional):

Reviewers' comments:

Reviewer's Responses to Questions

**Comments to the Author**

1. If the authors have adequately addressed your comments raised in a previous round of review and you feel that this manuscript is now acceptable for publication, you may indicate that here to bypass the “Comments to the Author” section, enter your conflict of interest statement in the “Confidential to Editor” section, and submit your "Accept" recommendation.

Reviewer #1: All comments have been addressed

Reviewer #2: All comments have been addressed

2. Is the manuscript technically sound, and do the data support the conclusions?

Reviewer #1: Yes

Reviewer #2: (No Response)

3. Has the statistical analysis been performed appropriately and rigorously? 

Reviewer #1: Yes

Reviewer #2: (No Response)

4. Have the authors made all data underlying the findings in their manuscript fully available?

Reviewer #1: Yes

Reviewer #2: (No Response)

5. Is the manuscript presented in an intelligible fashion and written in standard English?

Reviewer #1: Yes

Reviewer #2: (No Response)

6. Review Comments to the Author

Reviewer #1: ****************************************************

* All my comments have been addressed. I endorse publication *

Reviewer #2: (No Response)

7. PLOS authors have the option to publish the peer review history of their article (what does this mean?). If published, this will include your full peer review and any attached files.

Reviewer #1: No

Reviewer #2: No

---

## [Editor Report · Acceptance letter]

29 Oct 2021

PONE-D-21-23199R1 

Games and enculturation: A cross-cultural analysis of cooperative goal structures in Austronesian games  

Dear Dr. Leisterer-Peoples:

I'm pleased to inform you that your manuscript has been deemed suitable for publication in PLOS ONE. Congratulations! Your manuscript is now with our production department. 

Kind regards, 

on behalf of

Dr Pablo Brañas-Garza 

Academic Editor

PLOS ONE